# Plasma ACE2 predicts outcome of COVID-19 in hospitalized patients

Tue W. Kragstrup[1,2]*, Helene Søgaard Singh[1], Ida Grundberg[3], Ane Langkilde-Lauesen Nielsen[1,4], Felice Rivellese[5], Arnav Mehta[6,7], Marcia B. Goldberg[8,9], Michael R. Filbin[10], Per Qvist[1,11,12], Bo Martin Bibby[13]

**1** Department of Biomedicine, Aarhus University, Aarhus, Denmark, **2** Department of Rheumatology, Aarhus University Hospital, Aarhus, Denmark, **3** Olink Proteomics, Uppsala, Sweden, **4** Randers Regional Hospital, Randers, Denmark, **5** Centre for Experimental Medicine and Rheumatology, William Harvey Research Institute, Barts and The London School of Medicine and Dentistry, Queen Mary University of London, London, United Kingdom, **6** Department of Medical Oncology, Massachusetts General Hospital, Boston, Massachusetts, United States of America, **7** Broad Institute of MIT and Harvard, Cambridge, Massachusetts, United States of America, **8** Department of Medicine, Massachusetts General Hospital, Boston, Massachusetts, United States of America, **9** Department of Microbiology, Harvard Medical School, Boston, Massachusetts, United States of America, **10** Department of Emergency Medicine, Massachusetts General Hospital, Boston, Massachusetts, United States of America, **11** The Lundbeck Foundation Initiative for Integrative Psychiatric Research, iPSYCH, Aarhus, Denmark, **12** Centre for Genomics and Personalized Medicine, CGPM, Aarhus University, Aarhus, Denmark, **13** Department of Biostatistics, Aarhus University, Aarhus, Denmark

* kragstrup@biomed.au.dk

**Data Availability Statement:** Data is available from https://www.olink.com/mgh-covid-study/.

**Funding:** American Lung Association COVID-19 Action Initiative award (MBG). Independent Research Fund Denmark clinician scientist award

## Abstract

### Aims

Severe acute respiratory syndrome coronavirus 2 (SARS-CoV-2) binds to angiotensin converting enzyme 2 (ACE2) enabling entrance of the virus into cells and causing the infection termed coronavirus disease of 2019 (COVID-19). Here, we investigate associations between plasma ACE2 and outcome of COVID-19.

### Methods and results

This analysis used data from a large longitudinal study of 306 COVID-19 positive patients and 78 COVID-19 negative patients (MGH Emergency Department COVID-19 Cohort). Comprehensive clinical data were collected on this cohort, including 28-day outcomes. The samples were run on the Olink® Explore 1536 platform which includes measurement of the ACE2 protein. High admission plasma ACE2 in COVID-19 patients was associated with increased maximal illness severity within 28 days with OR = 1.8, 95%-CI: 1.4–2.3 ($P < 0.0001$). Plasma ACE2 was significantly higher in COVID-19 patients with hypertension compared with patients without hypertension ($P = 0.0045$). Circulating ACE2 was also significantly higher in COVID-19 patients with pre-existing heart conditions and kidney disease compared with patients without these pre-existing conditions ($P = 0.0363$ and $P = 0.0303$, respectively).

### Conclusion

This study suggests that measuring plasma ACE2 is potentially valuable in predicting COVID-19 outcomes. Further, ACE2 could be a link between COVID-19 illness severity and

(9039-00015B, TWK). Olink Proteomics financed and performed the proteomics assays presented in this work as part of the collaboration with Massachusetts General Hospital (MGH and the Broad Institute on the MGH Emergency Department COVID-19 Cohort. Olink Proteomics provided support in the form of salaries for IG, but did not have any additional role in the study design, data collection and analysis, decision to publish, or preparation of the manuscript. The specific roles of all authors are articulated in the 'author contributions' section.

**Competing interests:** IG is an employee of Olink Proteomics. This does not alter our adherence to PLOS ONE policies on sharing data and materials. The authors declare no other potential conflicts of interest.

**Abbreviations:** ACE, Angiotensin converting enzyme; Ang I, Angiotensin I; Ang II, Angiotensin II; ANOVA, Analysis of variance; AT1R, Type 1 angiotensin II receptor; AUC, Area under the curve; BMI, Body mass index; CoV, Coronavirus; COVID-19, Corona virus disease 2019; CRP, C-reactive protein; IL, Interleukin; NPX, Normalized Protein eXpression; PEA, Proximity extension assay; RAAS, Renin-angiotensin-aldosterone-system; ROC, receiver operating characteristic; SARS-CoV-2, Severe acute respiratory syndrome coronavirus 2; TNF, Tumor necrosis factor.

its established risk factors hypertension, pre-existing heart disease and pre-existing kidney disease.

## Introduction

Since December 2019, a previously undiscovered virus, severe acute respiratory syndrome coronavirus 2 (SARS-CoV-2), has caused a devastating global pandemic. The disease caused by SARS-CoV-2 infection has been termed coronavirus disease of 2019 (COVID-19) with clinical manifestations ranging from asymptomatic and subclinical infection to severe hyperinflammatory syndrome and death [1]. Risk factors for fatal infection are male gender, increased age and comorbidities including pre-existing hypertension, heart disease, lung disease, diabetes, kidney disease and immune suppression [2, 3].

SARS-CoV-2 binds to the angiotensin converting enzyme 2 (ACE2) receptor enabling entrance into cells through membrane fusion and endocytosis [4–8]. The ACE2-receptor is distributed in different tissues including vascular endothelial cells, smooth muscle cells, nasal and oral mucosa, enterocytes within the intestines, and is especially abundant in the kidneys [9, 10] and type II alveolar pneumocytes in the lungs [11, 12]. This distribution explains possible entry routes for the virus, and why target cells such as the pneumocytes are highly vulnerable to viral infection [12].

ACE2 is part of the renin-angiotensin-aldosterone-system (RAAS). Renin cleaves angiotensinogen leading to formation of angiotensin I (Ang I). Ang I is then converted to the vasoconstricting angiotensin II (Ang II) through cleavage by angiotensin-converting-enzyme (ACE), which is found in the vascular endothelium and more plentifully in the pulmonary endothelium [13, 14]. ACE2 opposes the effects of the RAAS by cleavage of Ang II into angiotensin (1–7), thereby attenuating increases in blood pressure [15]. Dysregulation of RAAS is therefore implicated in many diseases including hypertension and kidney disease [14, 16].

ACE2 is a tissue enzyme and thus circulating levels are low; the significance of measuring circulating ACE2 in pathologic conditions remains uncertain [17–19]. However, circulating ACE2 is elevated in patients with active COVID-19 disease and in the period after infection [20–23]. Further, elevated circulating ACE2 has been measured in patients with risk factors for severe COVID-19 disease. In patients with heart failure, plasma levels of ACE2 were higher in men compared with women and in patients with aortic stenosis [24, 25]. Increased circulating ACE2 has also been associated with increased risk of major cardiovascular events [26]. Finally, ACE2 levels were recently shown to be significantly elevated in serum from smokers, obese and diabetic individuals [27]. Therefore, strategies to use soluble recombinant ACE2 as a treatment in COVID-19 are being investigated [28].

This study is the first description of the association between circulating ACE2 and disease outcomes in patients with COVID-19 disease. Data were obtained from a large longitudinal COVID-19 study [29], with a publicly-available dataset including proteomics provided by Olink Proteomics.

## Methods

### Study populations

Data were downloaded from https://www.olink.com/mgh-covid-study/ on 14 Sep 2020. A detailed description is available online (https://www.olink.com/mgh-covid-study/) (Fig 1) and has been published previously [29]. Subjects included patients 18 years or older, who were in

acute respiratory distress with a clinical concern for COVID-19 upon arrival to the Emergency Department. Of the 384 patients enrolled, 306 (80%) tested positive and 78 tested negative for SARS-CoV-2. COVID-19-positive patients had blood samples drawn on days 0, 3, and 7, while virus-negative patients only had samples drawn on day 0. Clinical data collected included 28-day outcome classification based on the maximal illness severity (Acuity max, A1-A5) experience during the first 28 days after enrollment: A1 = Death. A2 = Intubated, ventilated, survived. A3 = Hospitalized, supplementary $O_2$ required. A4 = Hospitalized, no supplementary $O_2$ required. A5 = Discharged directly from ED and not subsequently hospitalized within 28 days. We also defined a dichotomous severity outcome with severe as A1-A2, and non-severe as A3-5. Discharged directly from ED and not subsequently hospitalized within 28 days [30]. Patient age, body mass index (BMI) and pre-existing medical conditions were recorded, including heart conditions (coronary artery disease, congestive heart failure, valvular disease), kidney disease (chronic kidney disease, baseline creatinine >1.5, end stage renal disease), lung disease (asthma, chronic obstructive lung disease, requiring home $O_2$, any chronic lung condition), diabetes (pre-diabetes, insulin and non-insulin dependent diabetes), and immunosuppressive conditions (active cancer, chemotherapy, transplant, immunosuppressive agents, asplenia). Laboratory tests used in the analyses of this study were C-reactive protein (CRP), absolute neutrophil count, and D-dimer. Continuous variables were categorized to make the dataset anonymized. Age (years) categories were 1 = 20–34, 2 = 35–49, 3 = 50–64, 4 = 65–79, 5 = 80+. BMI (kg/m$^2$) categories were 0 = <18.5 (underweight), 1 = 18.5–24.9 (normal), 2 = 25.0–29.9 (overweight), 3 = 30.0–39.9 (obese), 4 = 40+ (severely obese), 5 = Unknown (these were excluded from analysis on BMI). CRP (mg/L) categories were 1 = 0–19.9, 2 = 20–59.0, 3 = 60–99.9, 4 = 100–179, 5 = 180+. Absolute neutrophil count ($10^9$/L) categories were 1 = 0–0.99, 2 = 1.0–3.99, 3 = 4.0–7.99, 4 = 8.0–11.99, 5 = 12+. D-dimer (fibrinogen-equivalent units) categories were 1 = 0–499, 2 = 500–999, 3 = 1000–1999, 4 = 2000–3999, 5 = 4000+.

COVID-19-negative subjects enrolled were older than COVID-19-positive patients, less Hispanic, and with greater baseline burden of chronic illnesses. Of the 78 COVID-19-negative subjects, 37 (47%) were diagnosed with non-COVID-19 pneumonia or acute lung injury (e.g., aspiration), 12 (15%) with congestive heart failure exacerbation, 6 (7.7%) with COPD exacerbation, 3 (3.8%) with acute pulmonary embolus, 11 (14%) with non-pulmonary sepsis or infection, and 8 (10%) with other illnesses. COVID-19-negative patients were significantly less inflamed than COVID-19-positive patients, median CRP 22 [IQR 9–67] versus 105 [IQR 48–161], p-value < 0.05, but illness acuity and outcomes were very similar between the two groups [29].

## Ethics

Sample collection and analysis was approved by Partners Human Research Committee (PHRC). The need for informed consent was waived by this committee.

## Quantification of ACE2 and Olink data analysis

Detailed description is available online (https://www.olink.com/mgh-covid-study/) (Fig 1) and has been published previously [29]. Briefly, the samples were analyzed by the Olink® Explore 1536 platform which includes measurement of the ACE2 protein. The Olink platform is based on Proximity Extension Assay (PEA) technology and has been validated previously [31]. Data generation consists of three main steps: normalization to known standard (extension control), log2-transformation, and level adjustment using the plate control. The generated data represent relative protein values, Normalized Protein eXpression (NPX), on a log2 scale where a

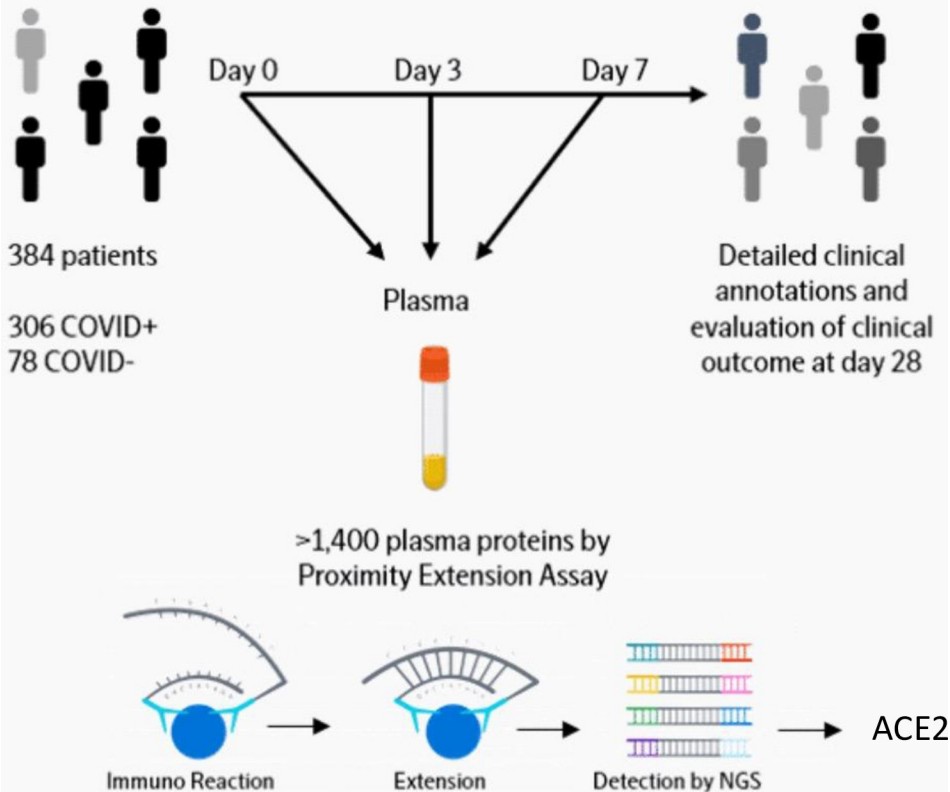

**Fig 1. Diagram of study work flow.** Link https://www.olink.com/mgh-covid-study/. Modified and used with permission.

larger number represents a higher protein level in the sample. For more information about Olink® Explore 1536, PEA and NPX, please visit www.olink.com.

## Statistical analyses

Group comparisons were made using non-parametric tests. The diagnostic value of plasma ACE2 at baseline was further tested with a receiver operating characteristic (ROC) curve for severe (A1-A2) vs non-severe (A3-A5). The comparison of outcome group differences between COVID-19 positive and negative patients was done using a two-way ANOVA. The association between clinical outcome (response variable) and plasma ACE2 (explanatory variable) was also investigated using ordered logistic regression and results are presented as odds ratios (OR) for worse outcome scores per unit increase in plasma ACE2. The specific statistical method tests used for each analysis is described in the table and figure legends. For all tests, the level of significance was a two-sided $P$ value of less than 0.05. Figures were made using Graph-Pad Prism 8 for Mac (GraphPad Software).

## Results

### Association between day 0 plasma ACE2 and worst clinical outcome group during the 28-day period in COVID-19 patients

First, we investigated the association between circulating ACE2 at day 0 and maximal acuity outcome group during the 28-day study period (Acuity max, A1-A5). Elevated baseline plasma ACE2

from COVID-19 patients was significantly associated with Acuity max with OR = 1.8, 95%-CI: 1.4–2.3 ($P < 0.0001$) (Table 1 and Fig 2A). The receiver operating characteristic (ROC) curve in Fig 2B depicts ability for ACE2 level at day 0 to discriminate severe (A1-A2) versus non-severe (A3-A5) outcome during the 28-day study period; the area under the curve (AUC) was 0.67, 95%-CI: 0.60–0.73. The association between circulating ACE2 at day 0 and Acuity max was also tested in regression models with correction for baseline characteristics, pre-existing medical conditions, and laboratory test results. In models correcting for age, body mass index, hypertension, and pre-existing heart conditions, kidney disease, lung disease, diabetes, and immunosuppressive conditions, significant associations were still observed between plasma ACE2 at day 0 and Acuity max (Table 1). Furthermore, we analyzed the association between plasma ACE2 at day 0 and Acuity max after correcting for CRP, absolute neutrophil count, and D-dimer to evaluate whether plasma ACE2 adds to the information already achieved by these laboratory test results. The association between elevated plasma ACE2 and Acuity max remained significant (Table 1).

## Association between plasma ACE2 at day 0, day 3, and day 7 and acuity group A2 (intubated) versus A3-A4 (not intubated) in hospitalized COVID-19 patients

Next, we tested whether circulating ACE2 at day 0, day 3, and day 7 was associated with clinical status at the time of blood sampling. Hospitalized patients were grouped according to outcome categories A2 (intubated at the time of sample collection) or A3-A4 (not intubated at the time of sample collection). Elevated plasma ACE2 in COVID-19 patients was significantly associated with higher acuity category at the time of blood sampling at day 0, day 3, and day 7 ($P = 0.0004$, $P < 0.0001$, and $P < 0.0001$, respectively) (Fig 3). We performed the same analysis in the patients with a plasma ACE2 measurement for all three time points (n = 130). In this analysis as well, plasma ACE2 in COVID-19 patients was significantly elevated in patients with outcome category A2 (intubated and survived) compared with outcome categories A3-A4 (not intubated) at the time of blood sampling at day 0, day 3, and day 7 ($P = 0.0016$, $P < 0.0001$, and $P < 0.0001$, respectively) (S1 Fig).

## Association between plasma ACE2 and comorbidities in COVID-19 patients

We then analyzed the relationship between ACE2 and comorbidities. Circulating ACE2 in COVID-19-positive patients with hypertension was significantly elevated compared with

**Table 1. Associations between day 0 plasma ACE2 and maximal acuity within 28 days in COVID-19 positive patients in ordered logistic regression models.**

| Model | n | OR | 95%-CI | p |
|---|---|---|---|---|
| 1 | 285 | 1.8 | 1.4–2.3 | <0.0001 |
| 2 | 285 | 1.9 | 1.4–2.4 | <0.0001 |
| 3 | 285 | 1.9 | 1.4–2.4 | <0.0001 |
| 4 | 265 | 1.4 | 1.1–1.9 | 0.007 |

The outcome is the maximum acuity score during the 28-day period with death (A1) being the highest acuity possible, and discharge without requiring admission within 28 days (A5) the least acuity possible.

Models (predictors): 1) Plasma ACE2. 2) Plasma ACE2, age, body mass index (BMI). 3) Plasma ACE2, age, body mass index (BMI), pre-existing hypertension, pre-existing heart disease, pre-existing lung disease, pre-existing kidney disease, pre-existing diabetes, pre-existing immunosuppressive condition. 4) Plasma ACE2, C-reactive protein (CRP), absolute neutrophile count, and D-dimer.

n = number of patients included in statistical analysis. OR = Odds ratio for a higher clinical outcome category per unit increase in plasma ACE2, 95%-CI = 95% confidence interval, p = p-value.

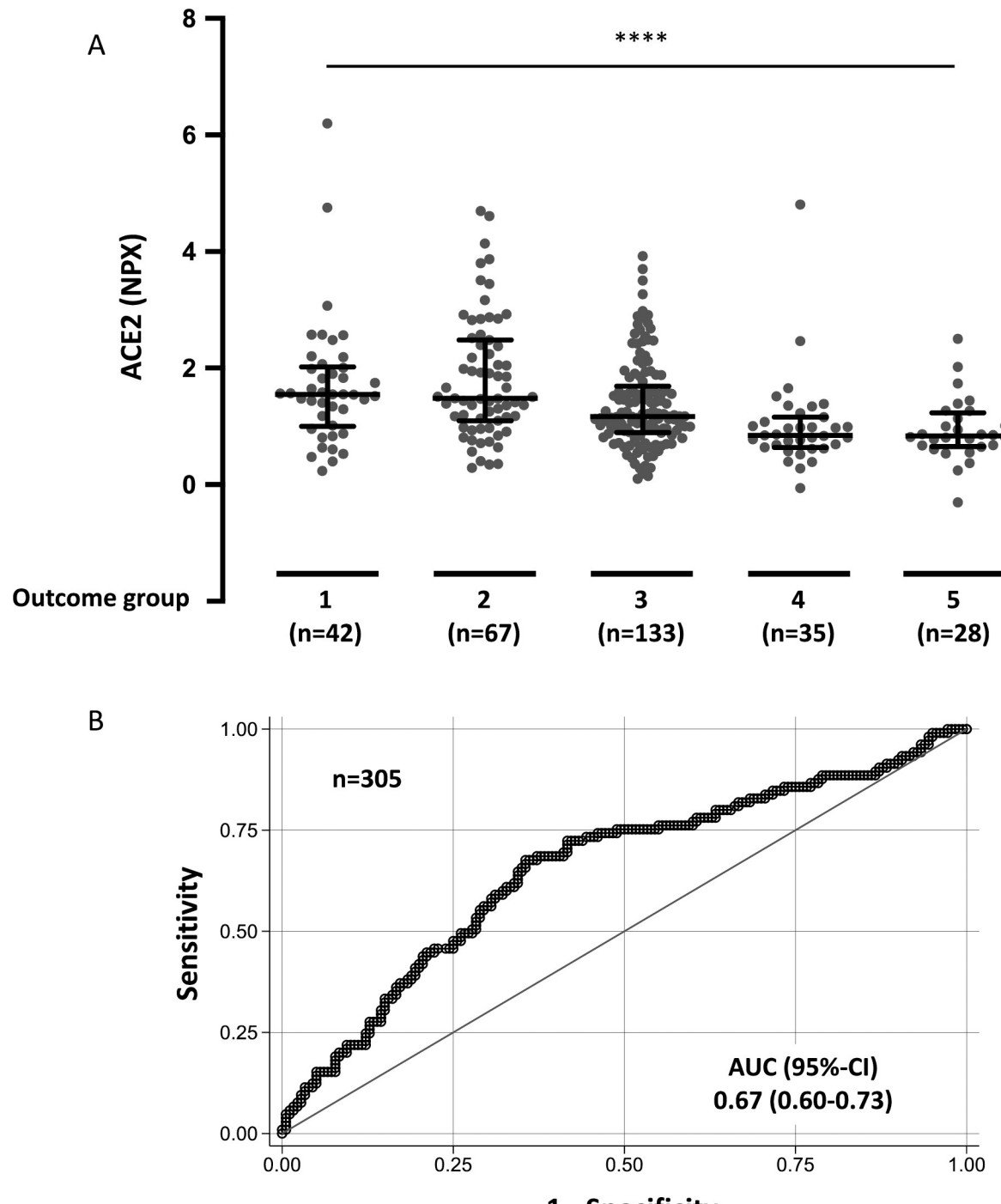

**Fig 2. Day 0 plasma ACE2 in COVID-19 positive patients by clinical outcome group and ROC curve. A.** Acuity max group is the maximum acuity score attained within the first 28 days with death being the maximum possible. A1 = Death within 28 days. A2 = Intubated, ventilated, survived to 28 days. A3 = Hospitalized, supplementary $O_2$ required. A4 = Hospitalized, no supplementary O2 required. A5 = Discharged directly from ED and not subsequently hospitalized within 28 days. Data were analyzed using the Kruskal-Wallis test. Bars indicate median and interquartile range. **** P < 0.0001. **B.** Receiver operating characteristic (ROC) curve comparing severe outcome groups A1-A2 vs non-severe outcome groups A3-A5 during the 28-day period.

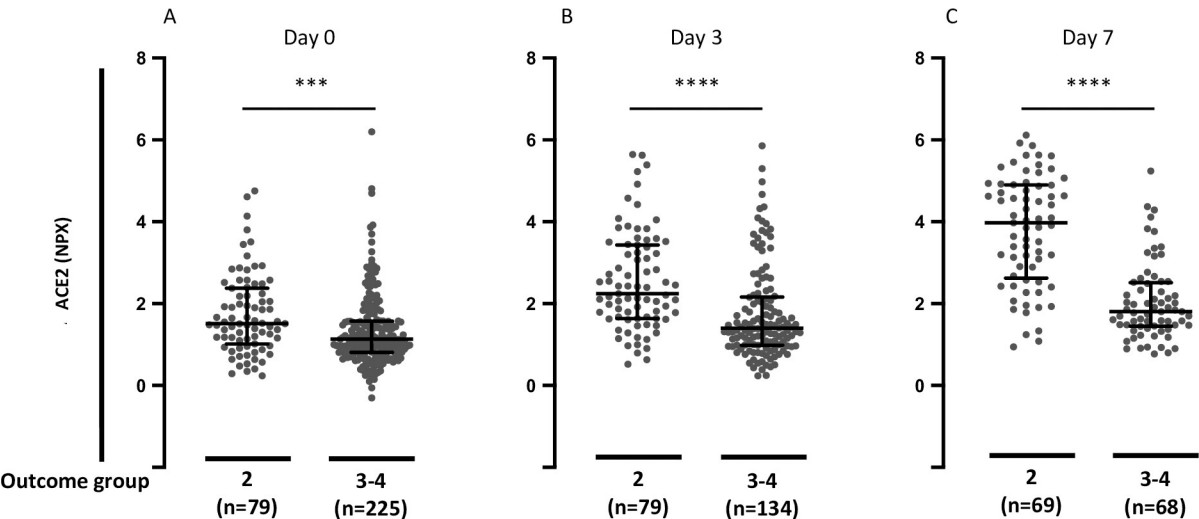

**Fig 3. Plasma ACE2 in hospitalized COVID-19 positive patients segregated by acuity groups A2 (intubated) versus A3-A4 (not intubated). A.** Day 0 plasma ACE2 in hospitalized COVID-19 positive patients by acuity groups A2 (intubated) or A3-A4 (not intubated) for day 0 study window (enrollment plus 24 hours). **B.** Day 3 plasma ACE2 in hospitalized COVID-19 positive patients by acuity groups A2 versus A3-A4 for day 3 study window. **C.** Day 7 plasma ACE2 in hospitalized COVID-19 positive patients by acuity groups A2 or A3-A4 for day 7 study window. Acuity categories: A2 = Intubated, ventilated. A3 = Hospitalized, supplementary O2 required. A4 = Hospitalized, no supplementary O2 required. Bars indicate median and interquartile range. Data were analyzed using the Mann Whitney test. *** P< 0.001. **** P < 0.0001.

patients without hypertension ($P = 0.0045$) (Fig 4). Circulating ACE2 was also significantly elevated in patients with pre-existing heart conditions and in patients with pre-existing kidney disease compared with patients without these pre-existing conditions ($P = 0.0363$ and $P = 0.0303$, respectively) (Fig 4). There was no significant difference in plasma ACE2 comparing patients with or without pre-existing lung disease, diabetes, or immunosuppressive conditions ($P = 0.953$, $P = 0.291$, and $P = 0.237$, respectively) (Fig 4).

## Association of plasma ACE2 with age and BMI in COVID-19 patients

We further tested associations of circulating ACE2 with age and body mass index (BMI). Elevated plasma ACE2 in COVID-19-positive patients was significantly associated with increasing age ($P = 0.0001$) (Fig 5). There was no significant association between plasma ACE2 and BMI ($P = 0.497$) (Fig 5).

## Association of day 0 plasma ACE2 with severe (A1-A2) versus non-severe (A3-A5) in COVID-19 positive and negative patients

Next, we analyzed whether circulating ACE2 differed between COVID-19 patients and non-COVID-19 patients with respiratory symptoms. Plasma ACE2 showed a clear overlap between the two groups. There was no significant difference between plasma ACE2 in COVID-19-positive versus negative patients ($P = 0.13$). Finally, we tested whether the association between plasma ACE2 and the maximal acuity score during the 28-day period was found in both COVID-19 positive and negative patients. The patients were grouped according to severe (A1-A2) or non-severe (A3-A5). Elevated plasma ACE2 in COVID-19-positive patients was significantly associated with maximal severity during the 28-day period ($P < 0.0001$), whereas there was no significant association between plasma ACE2 and outcome category in COVID-19 negative patients ($P = 0.085$) (Fig 6). To examine whether plasma ACE2 was differentially

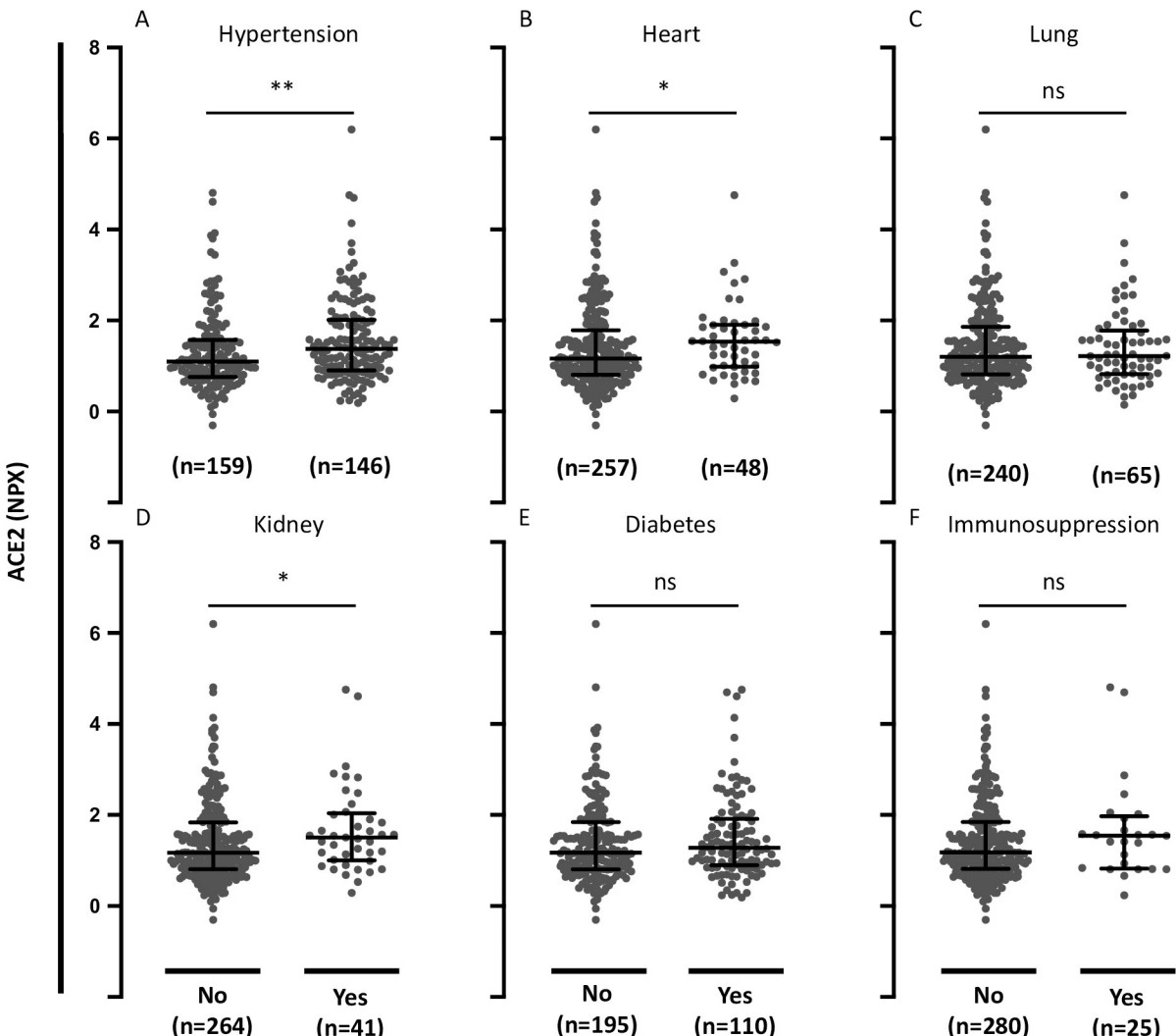

**Fig 4. Plasma ACE2 in COVID-19-positive patients with comorbidities. A.** Plasma ACE2 in COVID-19-positive patients with or without pre-existing hypertension. **B.** Plasma ACE2 in COVID-19-positive patients with or without pre-existing heart disease (coronary artery disease, congestive heart failure, valvular disease). **C.** Plasma ACE2 in COVID-19-positive patients with or without pre-existing lung disease (asthma, COPD, requiring home $O_2$, any chronic lung condition). **D.** Plasma ACE2 in COVID-19-positive patients with or without pre-existing kidney disease (chronic kidney disease, baseline creatinine >1.5, ESRD). **E.** Plasma ACE2 in COVID-19-positive patients with or without pre-existing diabetes (pre-diabetes, insulin and non-insulin dependent diabetes). **F.** Plasma ACE2 in COVID-19-positive patients with or without pre-existing immunocompromised condition (active cancer, chemotherapy, transplant, immunosuppressant agents, asplenic). Data were analyzed using the Mann Whitney test. Bars indicate median and interquartile range. * P< 0.05. ** P< 0.01. ns = not significant.

distributed among the clinical outcome groups in COVID-19-positive versus negative patients, we also conducted a two-way analysis of variance (ANOVA). Elevated plasma ACE2 was significantly associated with higher Acuity max during the 28-day period in COVID-19-positive patients and not in COVID-19-negative patients. However, differences were not statistically different (Table 2).

## Discussion

SARS-CoV-2 uses ACE2 as a functional receptor for entry into cells [32]. Circulating ACE2 is increased in patients with active COVID-19 infection compared with healthy controls [21–

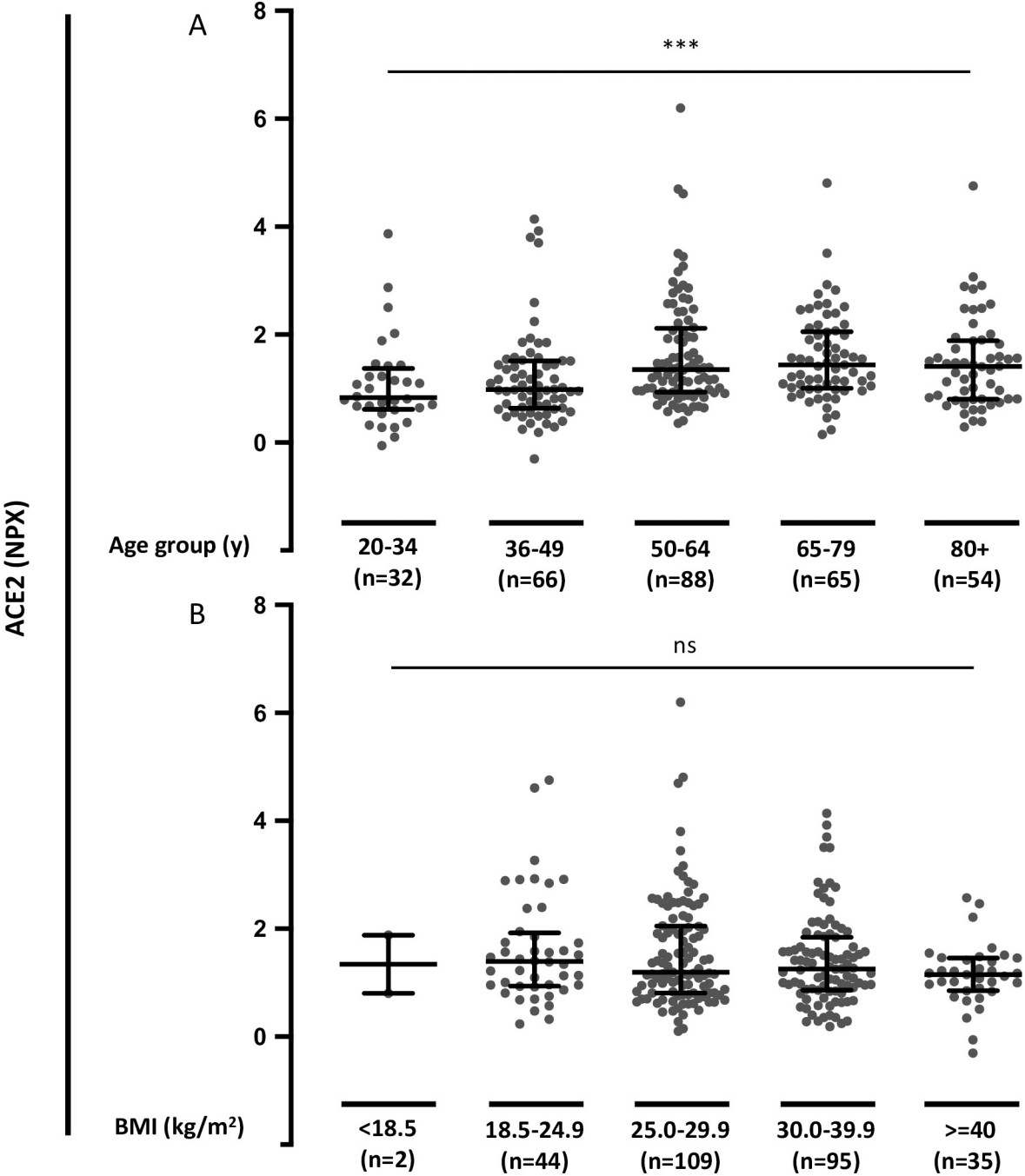

**Fig 5. Plasma ACE2 in COVID-19-positive patients in relation to age and BMI. A.** Plasma ACE2 in COVID-19-positive patients by age groups. **B.** Plasma ACE2 in COVID-19 positive patients by body mass index (BMI) groups. Data were analyzed using the Kruskal-Wallis test. *** P< 0.001. ns = not significant.

23]. Our analysis is the first description of circulating ACE2 in association with clinical outcomes in patients with COVID-19 disease. Data were obtained from a large longitudinal COVID-19 study with proteomics and clinical data, including disease severity evaluated through 28 days [29].

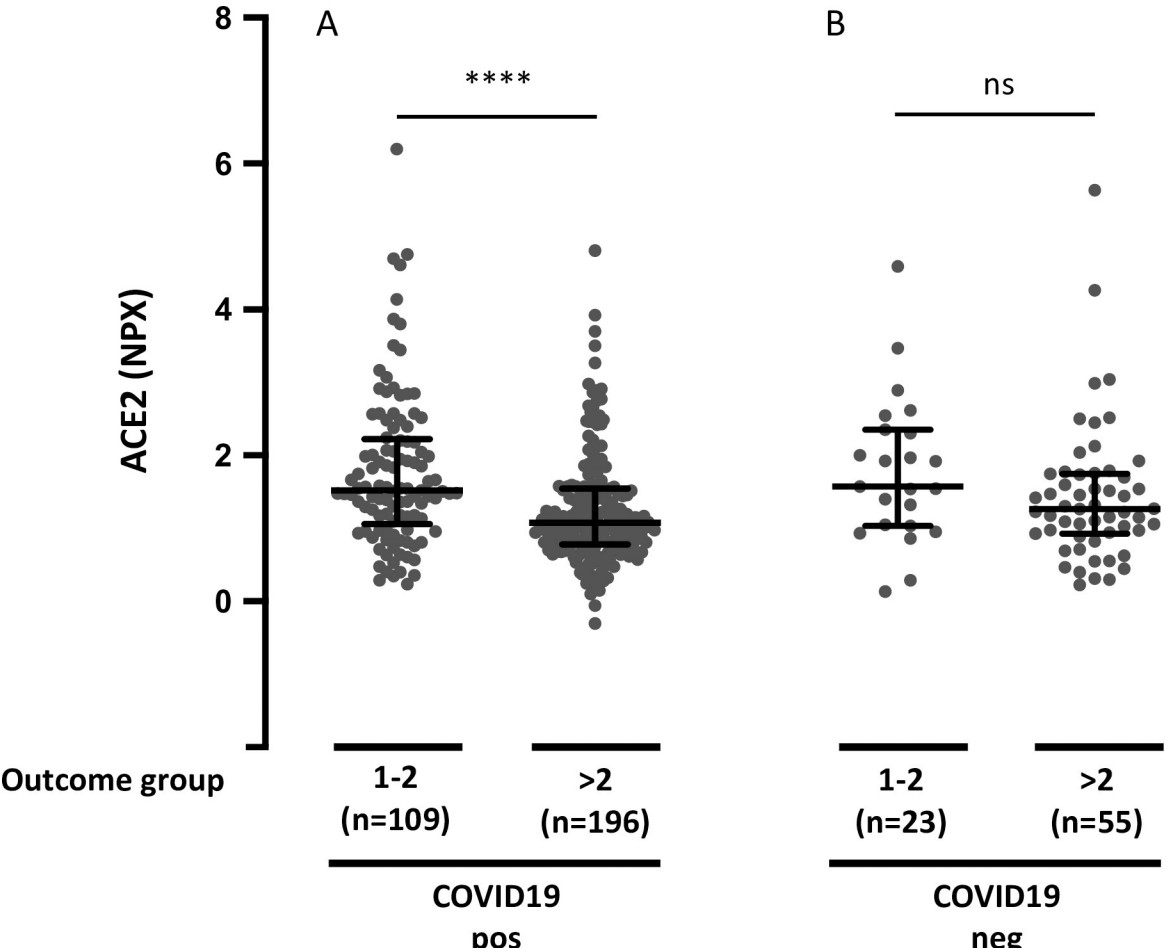

**Fig 6. Plasma ACE2 and maximal acuity group during the 28-day period in COVID-19 positive and negative patients. A.** Plasma ACE2 and outcome group in COVID-19-positive patients. **B.** Plasma ACE2 and outcome group in COVID-19 negative patients. The acuity group is the maximum acuity group during the 28-day period with death being the maximum possible. A1 = Death within 28 days. A2 = Intubated, ventilated, survived to 28 days. A3 = Hospitalized, supplementary O2 required. A4 = Hospitalized, no supplementary O2 required. A5 = Discharged directly from ED and not subsequently hospitalized within 28 days. Data were analyzed using the Mann Whitney test. Bars indicate median and interquartile range. **** $P < 0.0001$. ns = not significant.

Elevated baseline plasma ACE2 in COVID-19 patients was significantly associated with increased disease severity during the 28-day study period. This indicates that abundant ACE2 production could be involved in increased viral spread and disease burden, as previously

**Table 2. Associations between day 0 plasma ACE2 and maximal acuity group during the 28-day period (Acuity max) in COVID-19-positive and negative patients in two-way analysis of variance (ANOVA).**

|  | Outcome group A1-2 vs. A3-5 | p | 95%-CI |
| --- | --- | --- | --- |
| COVID-19-negative | 0.33 | 0.15 | -0.12–0.77 |
| COVID-19-positive | 0.49 | <0.0001 | 0.27–0.71 |
| Difference | 0.16 | 0.52 | -0.34–0.66 |

Maximal acuity category at 28 days is the maximum score within first 28 days with death (A1) being the highest acuity possible, and discharge without requiring admission within 28 days (A5) the least acuity possible.

p = p-value, 95%-CI = 95% confidence interval.

shown in experimental models of SARS-CoV infection [32, 33], with similar mechanisms postulated to be relevant for SARS-CoV-2 infection [34]. The association of elevated plasma ACE2 with higher clinical acuity and worse outcomes would in theory support blockade of ACE2 as a therapeutic strategy. ACE2 is expressed in the type II alveolar pneumocytes [12], which produce surfactant and act as progenitors for the type I alveolar pneumocytes [35]. The binding of the virus to ACE2 on the type II alveolar pneumocytes and subsequent infection leads to depletion of these cells, resulting in a decrease in the production and secretion of surfactant, as well as a lack of ability to regenerate and repair injured lung tissue, leading to the exacerbation of lung injury in severe COVID-19 [35, 36]. This interpretation is however not supported by findings in animal models, showing how ACE2 protects from severe acute lung injury induced by acid aspiration or sepsis [37]. After induction of severe acute lung injury through acid aspiration, ACE2-knockout mice had worsened oxygenation, massive pulmonary edema, and increased inflammatory cell infiltration compared to the wild type mice. In wild type mice, acid aspiration provoked a marked downregulation of ACE2 and an increase in Ang II levels without affecting ACE levels. In ACE2-knockout mice, Ang II levels increased to a greater extent, promoting further lung damage. Moreover, Ang II has been found to activate several cells within the immune system, such as macrophages, leading to a higher production of proinflammatory cytokines like IL-6 and TNFα [35, 36, 38]. Therefore, the direct blockade of ACE2, by removing the physiological brakes on the angiotensin system, could have detrimental effects on COVID-19 disease. For these reasons, an alternative strategy to inhibit the interaction of SARS-CoV-2 with ACE2 could be the administration of recombinant soluble ACE2 as a decoy receptor, which has been previously tested in small clinical trials in acute respiratory distress syndrome [39] and is now being explored in COVID-19 disease [40–42]. However, the level of circulating ACE2 does not necessarily reflect the expression of ACE2 in the plasma membrane of host cells. Thus, high levels of ACE2 in the plasma might result from increased lysis of ACE2-expressing cells as a consequence of a more severe infection.

It is desirable to predict disease outcomes in order to specialize treatment, since the drugs used in the treatment of COVID-19 can have deleterious side effects [43]. The ROC curve showed moderate value of baseline ACE2 levels for discriminating patients with severe (death or intubation) vs non-severe disease within 28 days. Further, difference in median plasma ACE2 levels between groups A2 (intubated, survived) versus A3-A4 (hospitalized but not intubated) increased over time on day 3 and day 7 of admission. Therefore, repeated analysis of plasma ACE2 may provide increased predictive value. This is in line with a recent study showing increasing SARS-CoV-2 viral load during hospitalization and prolonged viral shedding in more severe COVID-19 disease [44]. Further, it is interesting that the association between plasma ACE2 and maximal acuity within 28 days in COVID-19 patients was statistically significant also after correction for age, BMI, pre-existing medical conditions, and the laboratory tests CRP, absolute neutrophil count and D-dimer. This suggests that measuring plasma ACE2 adds to the value of clinically available data to help predict disease outcome in COVID-19. ACE2 is a membrane-bound enzyme, and therefore measuring the circulating and urine levels of ACE2 is complex [38]. Soluble ACE2 is the result of the cleavage of the membrane-bound ACE2 by disintegrin and metalloprotease 17. Interestingly, there is a correlation between plasma ACE1 activity and ACE2 activity in healthy individuals [45]. If this same correlation is seen in patients with COVID-19, plasma ACE1 levels (serum ACE) could be used to approximate the activity of ACE2 in COVID-19 positive patients. This is noteworthy because serum ACE analysis is a standardized test in most international hospitals.

Circulating ACE2 in COVID-19-positive patients with hypertension was significantly increased compared with plasma from patients without hypertension. Patients with hypertension are often treated with ACE-inhibitors and AT1R-blockers. During this pandemic, it has

been highly debated whether or not the use of these antihypertensive medications should be discontinued in patients with COVID-19. A study performed on rats showed that the use of ACE inhibitors and/or use of AT1R-blocker led to an increased expression of ACE2 in cardiac tissue [46]. Others found a decrease of kidney ACE2 expression and no effect on lung ACE2 expression with ACE-inhibitors and AT1R-blockers [47]. These findings have raised the concern that patients with hypertension, who are treated with ACE2-modulating drugs like ACE-inhibitors or AT1R-blockers might be at a higher risk for severe COVID-19 infection, since it could alter the entry-way for the virus [48]. A recent study showed that plasma ACE2 activity is increased in COVID-19 patients treated with ACE inhibitors [49]. However, more severe COVID-19 disease in patients treated with ACE-inhibitors or AT1R-blockers has not been supported by recent population based data [50]. Circulating ACE2 was also significantly increased in patients with pre-existing heart conditions and in patients with pre-existing kidney disease. This is in line with the central role of RAAS in these conditions. Importantly, the changes in circulating ACE2 in COVID-19 disease observed here are small and are likely not to be of any significance regarding the metabolism of the substrates of ACE2.

In contrast, differences in plasma ACE2 does not seem to explain the risk of severe COVID-19 disease associated with pre-existing lung disease, diabetes, or immunosuppression. Plasma ACE2 was associated with age, which is in line with recent observations showing higher serum levels of ACE2 in adults compared to a pediatric cohort [51]. There were no associations between plasma ACE2 and BMI.

Baseline plasma ACE2 was not significantly different between COVID-19-positive and negative patients, which is in line with the results of a recent study with a much smaller sample size [52]. The association between elevated plasma ACE2 and maximal acuity within 28 days were more pronounced in COVID-19-positive patients compared with COVID-19-negative patients, but the difference was not significant in a two-way ANOVA analysis. It is therefore not possible with this dataset to conclude that the observed association between circulating ACE2 and disease outcome is specific to patients with COVID-19-positive respiratory disease.

There are some overall limitations of this study. First, plasma ACE2 was measured as relative protein concentrations using NPX (Normalized Protein eXpression) values. Therefore, it is not possible to determine a plasma ACE2 cut-off value to predict severe outcome or to compare findings in this study with results in studies measuring protein concentration or enzymatic activity. Second, several continuous variables were categorized in the publicly available data set. This decreases the statistical power of some of the analysis. Third, some potentially important variables such as gender and treatment were not available. Fourth, severity driven recruitment criteria (respiratory distress at emergency department) might have introduced a bias. Thus, patients with more severe COVID-19 at admission have both higher ACE2 at time of inclusion and a poorer prognosis. However, our study reflects clinical practice where laboratory tests are performed when patients are admitted to the hospital. Finally, causality of associations between plasma ACE2 and severity of COVID-19 disease cannot be drawn from this study.

## Conclusions

Overall, this study suggests a potential utility of measuring ACE2 in COVID-19 to predict disease outcome. Further, circulating ACE2 could be a link between severe COVID-19 disease and its risk factors, namely hypertension, pre-existing heart disease and pre-existing kidney disease. The design of the data analysis using the Olink platform does not allow assessment of quantitative differences. However, previous studies have described a positive correlation between plasma ACE2 and ACE1 activity. This is interesting because ACE1 (serum ACE)

analysis is a standardized test in most hospital laboratories. Therefore, our study encourages quantitative investigations of both plasma ACE 1 and 2 in COVID-19.

## Supporting information

**S1 Fig. Plasma ACE2 in hospitalized COVID-19 positive patients with samples analyzed at all time points divided in clinical outcome groups 2 (intubated) or 3–4 (not intubated). A.** Day 0 plasma ACE2 in hospitalized COVID-19 positive patients divided in outcome groups 2 (intubated) or 3–4 (not intubated) for day 0 study window (enrollment plus 24 hours). **B.** Day 3 plasma abundance of ACE2 in hospitalized COVID-19 positive patients divided in outcome groups 2 (intubated) or 3–4 (not intubated) for day 3 study window. **C.** Day 7 plasma ACE2 in hospitalized COVID-19 positive patients divided in outcome groups 2 (intubated) or 3–4 (not intubated) for day 7 study window. The acuity groups: 2 = Intubated, ventilated. 3 = Hospitalized, supplementary O2 required. 4 = Hospitalized, no supplementary O2 required. Bars indicate median and interquartile range. Data were analyzed using the Mann Whitney test. ** $P <$ 0.01. **** $P < 0.0001$.
(DOCX)

## Acknowledgments

Data provided by the MGH Emergency Department COVID-19 Cohort (Filbin, Goldberg, Hacohen) with Olink Proteomics. We want to thank Aparna Udupi, Department of Biostatistics, Aarhus University for technical assistance running the statistical analyses.

## Author Contributions

**Conceptualization:** Tue W. Kragstrup, Ida Grundberg, Arnav Mehta, Marcia B. Goldberg, Michael R. Filbin, Per Qvist, Bo Martin Bibby.

**Data curation:** Ida Grundberg, Arnav Mehta, Marcia B. Goldberg, Michael R. Filbin, Per Qvist.

**Formal analysis:** Tue W. Kragstrup, Ane Langkilde-Lauesen Nielsen, Arnav Mehta, Per Qvist, Bo Martin Bibby.

**Funding acquisition:** Marcia B. Goldberg.

**Investigation:** Helene Søgaard Singh, Ida Grundberg, Felice Rivellese, Marcia B. Goldberg, Michael R. Filbin, Per Qvist, Bo Martin Bibby.

**Methodology:** Tue W. Kragstrup, Ida Grundberg, Ane Langkilde-Lauesen Nielsen.

**Writing – original draft:** Tue W. Kragstrup, Helene Søgaard Singh.

**Writing – review & editing:** Tue W. Kragstrup, Helene Søgaard Singh, Ida Grundberg, Ane Langkilde-Lauesen Nielsen, Felice Rivellese, Arnav Mehta, Marcia B. Goldberg, Michael R. Filbin, Per Qvist, Bo Martin Bibby.

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
