## [Decision Letter · Decision Letter 0]

30 Mar 2021

PONE-D-21-06434

Plasma ACE2 levels predict outcome of COVID-19 in hospitalized patients

PLOS ONE

Dear Dr. Kragstrup,

Thank you for submitting your manuscript to PLOS ONE. After careful consideration, we feel that it has merit but does not fully meet PLOS ONE’s publication criteria as it currently stands. Therefore, we invite you to submit a revised version of the manuscript that addresses the points raised during the review process.

We look forward to receiving your revised manuscript.

Kind regards,

Michael Bader

Academic Editor

PLOS ONE

Journal Requirements:

"Patients were enrolled in the Emergency Department of a large, urban, academic hospital from 3/24/2020 to 4/30/2020 in Boston during the peak of the COVID-19 surge, with an institutional IRB-approved waiver of informed consent."

'IG is an employee of Olink Proteomics. The authors declare no other potential conflicts of interest.'

We note that one or more of the authors have an affiliation to the commercial funders of this research study : Olink Proteomics.

5. We note that Figure 1 in your submission contains copyrighted images.

All PLOS content is published under the Creative Commons Attribution License (CC BY 4.0), which means that the manuscript, images, and Supporting Information files will be freely available online, and any third party is permitted to access, download, copy, distribute, and use these materials in any way, even commercially, with proper attribution. For more information, see our copyright guidelines: http://journals.plos.org/plosone/s/licenses-and-copyright.

We require you to either (a) present written permission from the copyright holder to publish this figure specifically under the CC BY 4.0 license, or (b) remove the figure from your submission:

b.  If you are unable to obtain permission from the original copyright holder to publish this figure under the CC BY 4.0 license or if the copyright holder’s requirements are incompatible with the CC BY 4.0 license, please either i) remove the figure or ii) supply a replacement figure that complies with the CC BY 4.0 license. Please check copyright information on all replacement figures and update the figure caption with source information. If applicable, please specify in the figure caption text when a figure is similar but not identical to the original image and is therefore for illustrative purposes only.

6. Please amend your list of authors on the manuscript to ensure that each author is linked to an affiliation. Authors’ affiliations should reflect the institution where the work was done (if authors moved subsequently, you can also list the new affiliation stating “current affiliation:….” as necessary).

Reviewers' comments:

Reviewer's Responses to Questions

**Comments to the Author**

1. Is the manuscript technically sound, and do the data support the conclusions?

Reviewer #1: No

Reviewer #2: Yes

2. Has the statistical analysis been performed appropriately and rigorously? 

Reviewer #1: I Don't Know

Reviewer #2: Yes

3. Have the authors made all data underlying the findings in their manuscript fully available?

Reviewer #1: No

Reviewer #2: Yes

4. Is the manuscript presented in an intelligible fashion and written in standard English?

Reviewer #1: Yes

Reviewer #2: Yes

5. Review Comments to the Author

Reviewer #1: In the current paper, the authors investigate the link of ACE2 signals obtained from a proximal extension assay multiplex panel in a cohort of 306 COVID-19 patients and 78 SARS-CoV-2 negative controls. The authors suggest a link between baseline signals for ACE2 obtained from a commercially available proximal extension assay and the outcome of COVID-19 disease. The findings are interesting and in-line with previous observations regarding ACE2 regulation during COVID-19. The authors should also check for more recent literature on the topic, describing a link between soluble ACE2 activity and COVID-19 severity.

The aspect of a prognostic value of baseline ACE2 for COVID-19 outcome appears to be interesting at the first glance, but considering the design of the clinical readout, it might be introduced by the fact that "Day 0" in the current study was actually the admission to the emergency department respiratory distress, which could happen at very variable time points related to disease onset. Knowing that COVID-19 severity is linked to plasma ACE2 activity with a peak in concentrations between 1 and 2 weeks after disease onset, severity driven recruitment criteria (respiratory distress at emergency department) might have introduces a certain bias, as more severe cases have higher ACE2 at time of inclusion, with obviously having poorer outcomes.

In other words, patients being included with a more severe disease manifestation ion day 0 (e.g. higher ACE2), may just be at another time-point in the course of COVID-19 disease, with an already pre-determined worse outcome. It would add a lot of valuer to the study, if the measurement time points would be related to the time of symptom onset, first positive test result or any other earlier time point that may would assure the direct comparability of disease severity groups. The authors should further describe the control group of 78 Covid-negative patients described more clearly. Why did these patients at emergency units with respiratory distress?

Finally, the use of terminology within the whole paper is misleading, as the authors keep using the terms "ACE2 levels" and "ACE2 concentrations". Of note, the result of the proximal extension assay the authors used as a basis for their interpretation is given in the form of an artificial unit (NPX), that has been calculated by the manufacturer by arithmetically linking a series of Ct value based correction algorithms. Even the manufacturers point out on their webpage that the given readout cannot be compared to actual protein levels, which disables comparability to other studies.

From a technical perspective, the big open question is why no calibration of the readout is performed to be able to provide actual protein levels. With a highly reproducible and standardized method as described by the manufacturers on their webpage, it should be easy to retrospectively include a valid calibration allowing for providing actual ACE2 concentrations instead of manufacturer invented units that prevent comparability with other studies. Moreover, a certain analytical validation vor ACE2 in the used PEA panel should be shown that compares the used "NPX" values with real concentration units or standard activity units in a defined set of clinical samples. It should also be noted that reading the method section needs some revision as it currently reads like an advertisement for the used (commercially available) technology rather than an objective method description to be published in a research paper.

Reviewer #2: .This study investigated the association between plasma ACE2 levels and outcomes of COVID-19 patients, using clinical data and plasma samples from 306 COVID-19 positive and 76 COVID-19 negative patients. High baseline plasma ACE2 levels are reported to be associated with worse COVID-19 outcomes and patients with hypertension, pre-existing heart conditions or kidney disease had higher plasma ACE2 levels than those without.

Even as a marker I doubt very much that it will be useful for COVID-19 because the changes found in figure 2 are so small.

Main criticisms and suggestions for improvement:

- Overall, the paper is an excellent contribution but the authors should acknowledge the limitations of measurements of ACE2 in plasma where the levels are usually very low and even when mildly elevated in pathological conditions the significance remains uncertain. Specifically, it must be stated that ACE2 is a tissue enzyme and that the levels in the circulation are low in all species studied, including humans. Appropriate references should be given.

Please acknowledge that the changes in COVID-19 are so small that they are not likely to be of any significance regarding the metabolism of the substrates of ACE2

- In the introduction the part on ACE2 receptor distribution must be modified to include the kidney as a main site of ACE2 . The authors do not seem aware of a critical important observation, namely that the expression of ACE2 in the lung is very low as shown by Serfozo et al using western blot and confirmed by others

See https://doi.org/10.1161/HYPERTENSIONAHA.119.14071 and https://doi.org/10.1038/s41467-020-19145-6

These references must be cited as well as others showing that the kidney is next to the intestine the organ that has the highest abundance of ACE2.

- Perhaps the authors are not aware that ACE2 RNA levels do not necessarily imply protein levels. Actually, ACE2 can be post translationally regulated and be increased when the levels of mRNA are not. The authors may consult and cite a review on this by Lores et al. https://doi.org/10.1042/CS20200484

- In the discussion it is stated that “ACE2 as a decoy receptor… is now being explored in COVID-19 disease.” Appropriate references should be given to be faithful to the literature. Perhaps they are not aware of these papers because they are recent but they should be cited. For instance, https://doi.org/10.1042/CS20200163 and https://doi.org/10.1681/ASN.2020101537

- “A study performed on rats showed that the use of ACE inhibitors and/or use of AT1R-blocker led to an increased expression of ACE2”. It should be specified that in ref 34 ACE2 was only examined in cardiac tissue and not kidney or lung. Others found a decrease of kidney ACE2 after ACEi and ARBs and no effect on lungs. For instance, see https://doi.org/10.1681/ASN.2020050667

- “The population sampled on day 3 and day 7 therefore consists of patients with more severe disease compared with day 0.” Maybe ACE2 for day 3 and day 7 should be shown in a way that they are only compared to their matching samples of day 0?

6. PLOS authors have the option to publish the peer review history of their article (what does this mean?). If published, this will include your full peer review and any attached files.

Reviewer #1: No

Reviewer #2: No

---

## [Author Response · Author response to Decision Letter 0]

20 May 2021

Response letter

Dear editor and reviewers,

We are very pleased with a constructive review of our paper. We readily acknowledge that reviewers have in depth knowledge on ACE2 physiology and the role of this protein in COVID-19 disease. We are therefore very happy to accommodate all suggestions made by the reviewers.

Three native speaking authors have carefully proof read the manuscript. This included a few changes worth mentioning.

1) The Odds ratios have been inverted to report odds ratios for a smaller category instead of a larger category. Seems like it would make more intuitive sense to have higher severity as the primary outcome in analyses. The text as written supports this approach.

2) Fig 2B has been changed to better align with the other figures (no colors).

We have made two general corrections throughout the manuscript based on reviewer suggestions and co-author comments after seeing the review. These changes are not highlighted in the manuscript because the corrections are found abundantly throughout.

1) As suggested by reviewer #1 we have replaced all mentions of ”levels” and ”concentrations” with ”plasma ACE2” or ”circulating ACE2”. 

2) We have also systematically changed the terminology of the disease groups from “WHO groups” to “clinical outcome groups/Acuity groups”. “WHO groups” was part of the publicly available information downloaded from the source link at the Olink website. However, the scores were not based on the WHO Outcome scale. This is because COVID19 is not treated with non-invasive ventilation or high-flow nasal cannula (therefore, no patients were categorized in previous group 3). The group definitions are the same but have been renamed accordingly. 1 = Death. 2 = Intubated, ventilated, survived. 3 = Hospitalized, supplementary O2 required. 4 = Hospitalized, no supplementary O2 required. 5 = Not hospitalized. These changes are not highlighted in the manuscript.

All other changes have been highlighted in the revised manuscript.

We believe the paper has been greatly improved. Thank you.

Tue Wenzel Kragstrup

Tue Wenzel Kragstrup

Associate Professor, MD, PhD

E-mail: kragstrup@biomed.au.dk

Biomedicine, Health

Aarhus University

Wilhelm Meyers Alle 4, 4fl.

DK-8000 Aarhus C

Response: We have made all formatting according to PLOS ONE's style requirements.

"Patients were enrolled in the Emergency Department of a large, urban, academic hospital from 3/24/2020 to 4/30/2020 in Boston during the peak of the COVID-19 surge, with an institutional IRB-approved waiver of informed consent."

Response: We have included full name of ethics committee in manuscript (Partners Human Research Committee).

Response: We have included full name of ethics committee in submission form (Partners Human Research Committee).

Response: Three native speaking authors have carefully proof read the manuscript (AM, MBG, and MF). Both reviewers find the manuscript presented in an intelligible fashion and written in standard English.

Response: We uploaded version with changes highlighted.

Response: We uploaded clean version.

'IG is an employee of Olink Proteomics. The authors declare no other potential conflicts of interest.'

We note that one or more of the authors have an affiliation to the commercial funders of this research study : Olink Proteomics.

Response: We have included the below statement about the role of Olink Proteomics. 

”Olink Proteomics financed and performed the proteomics assays presented in this work as part of the collaboration with Massachusetts General Hospital (MGH and the Broad Institute on the MGH Emergency Department COVID-19 Cohort.”

We already stated that IG (Ida Grundberg) is an employee of Olink. Now, as per your request we also included the statement below.

”Olink Proteomics provided support in the form of salaries for IG, but did not have any additional role in the study design, data collection and analysis, decision to publish, or preparation of the manuscript. The specific roles of all authors are articulated in the ‘author contributions’ section”

Response: We already stated that IG (Ida Grundberg) is an employee of Olink. Now, as per your request we also included the statement below.

”This does not alter our adherence to PLOS ONE policies on sharing data and materials”.

Response: We included Funding Statement and Competing Interests Statement in the updated Cover letter.

5. We note that Figure 1 in your submission contains copyrighted images.

All PLOS content is published under the Creative Commons Attribution License (CC BY 4.0), which means that the manuscript, images, and Supporting Information files will be freely available online, and any third party is permitted to access, download, copy, distribute, and use these materials in any way, even commercially, with proper attribution. For more information, see our copyright guidelines: http://journals.plos.org/plosone/s/licenses-and-copyright.

We require you to either (a) present written permission from the copyright holder to publish this figure specifically under the CC BY 4.0 license, or (b) remove the figure from your submission:

Response: We have uploaded written permission to publish this figure specifically under the CC BY 4.0 license.

b. If you are unable to obtain permission from the original copyright holder to publish this figure under the CC BY 4.0 license or if the copyright holder’s requirements are incompatible with the CC BY 4.0 license, please either i) remove the figure or ii) supply a replacement figure that complies with the CC BY 4.0 license. Please check copyright information on all replacement figures and update the figure caption with source information. If applicable, please specify in the figure caption text when a figure is similar but not identical to the original image and is therefore for illustrative purposes only.

6. Please amend your list of authors on the manuscript to ensure that each author is linked to an affiliation. Authors’ affiliations should reflect the institution where the work was done (if authors moved subsequently, you can also list the new affiliation stating “current affiliation:….” as necessary).

Response: The author affiliations are correct.

Reviewers' comments:

Reviewer's Responses to Questions

Comments to the Author

1. Is the manuscript technically sound, and do the data support the conclusions?

Reviewer #1: No

Reviewer #2: Yes

2. Has the statistical analysis been performed appropriately and rigorously? 

Reviewer #1: I Don't Know

Reviewer #2: Yes

3. Have the authors made all data underlying the findings in their manuscript fully available?

Response: The data is fully available without restriction and can be downloaded from https://www.olink.com/mgh-covid-study/. This has been highlighted in the Methods section.

Reviewer #1: No

Reviewer #2: Yes

4. Is the manuscript presented in an intelligible fashion and written in standard English?

Reviewer #1: Yes

Reviewer #2: Yes

5. Review Comments to the Author

Reviewer #1: In the current paper, the authors investigate the link of ACE2 signals obtained from a proximal extension assay multiplex panel in a cohort of 306 COVID-19 patients and 78 SARS-CoV-2 negative controls. The authors suggest a link between baseline signals for ACE2 obtained from a commercially available proximal extension assay and the outcome of COVID-19 disease. The findings are interesting and in-line with previous observations regarding ACE2 regulation during COVID-19. The authors should also check for more recent literature on the topic, describing a link between soluble ACE2 activity and COVID-19 severity. 

Response: We would like to thank the Reviewer for taking time to carefully read our paper. We agree with all his/her comments. We believe that we have addressed all comments and this is a great improvement to the manucript.

We reviewed the literature again and found the very recent publications below evaluating circulating ACE2 in COVID-19. They have been cited in the manuscript.

Lundström A, Ziegler L, Havervall S, Rudberg A-S, von Meijenfeldt F, Lisman T, et al. Soluble angiotensin-converting enzyme 2 is transiently elevated in COVID-19 and correlates with specific inflammatory and endothelial markers. medRxiv. 2021:2021.03.03.21252841. doi: 10.1101/2021.03.03.21252841.

Nagy B, Jr., Fejes Z, Szentkereszty Z, Suto R, Varkonyi I, Ajzner E, et al. A dramatic rise in serum ACE2 activity in a critically ill COVID-19 patient. Int J Infect Dis. 2021;103:412-4. Epub 2020/11/30. doi: 10.1016/j.ijid.2020.11.184. PubMed PMID: 33249290; 

van Lier D, Kox M, Santos K, van der Hoeven H, Pillay J, Pickkers P. Increased blood angiotensin converting enzyme 2 activity in critically ill COVID-19 patients. ERJ Open Res. 2021;7(1). Epub 2021/03/20. doi: 10.1183/23120541.00848-2020. 

Fagyas M, Kertesz A, Siket IM, Banhegyi V, Kracsko B, Szegedi A, et al. Level of the SARS-CoV-2 receptor ACE2 activity is highly elevated in old-aged patients with aortic stenosis: implications for ACE2 as a biomarker for the severity of COVID-19. Geroscience. 2021. Epub 2021/01/21. doi: 10.1007/s11357-020-00300-2. 

And per request from reviewer #2 we added these publications better refelcting the current litterature:

Lores E, Wysocki J, Batlle D. ACE2, the kidney and the emergence of COVID-19 two decades after ACE2 discovery. Clin Sci (Lond). 2020;134(21):2791-2805.

Lee IT, Nakayama T, Wu CT, et al. ACE2 localizes to the respiratory cilia and is not increased by ACE inhibitors or ARBs. Nat Commun. 2020;11(1):5453.

Serfozo P, Wysocki J, Gulua G, et al. Ang II (Angiotensin II) Conversion to Angiotensin-(1-7) in the Circulation Is POP (Prolyloligopeptidase)-Dependent and ACE2 (Angiotensin-Converting Enzyme 2)-Independent. Hypertension. 2020;75(1):173-182.

Batlle D, Wysocki J, Satchell K. Soluble angiotensin-converting enzyme 2: a potential approach for coronavirus infection therapy? Clin Sci (Lond). 2020;134(5):543-545.

Wysocki J, Ye M, Hassler L, et al. A Novel Soluble ACE2 Variant with Prolonged Duration of Action Neutralizes SARS-CoV-2 Infection in Human Kidney Organoids. J Am Soc Nephrol. 2021.

Epelman S, Shrestha K, Troughton RW, et al. Soluble angiotensin-converting enzyme 2 in human heart failure: relation with myocardial function and clinical outcomes. J Card Fail. 2009;15(7):565-571.

Tikellis C, Bialkowski K, Pete J, et al. ACE2 deficiency modifies renoprotection afforded by ACE inhibition in experimental diabetes. Diabetes. 2008;57(4):1018-1025.

Yamaleyeva LM, Gilliam-Davis S, Almeida I, Brosnihan KB, Lindsey SH, Chappell MC. Differential regulation of circulating and renal ACE2 and ACE in hypertensive mRen2.Lewis rats with early-onset diabetes. Am J Physiol Renal Physiol. 2012;302(11):F1374-1384.

Wysocki J, Lores E, Ye M, Soler MJ, Batlle D. Kidney and Lung ACE2 Expression after an ACE Inhibitor or an Ang II Receptor Blocker: Implications for COVID-19. J Am Soc Nephrol. 2020;31(9):1941-1943.

The aspect of a prognostic value of baseline ACE2 for COVID-19 outcome appears to be interesting at the first glance, but considering the design of the clinical readout, it might be introduced by the fact that "Day 0" in the current study was actually the admission to the emergency department respiratory distress, which could happen at very variable time points related to disease onset. Knowing that COVID-19 severity is linked to plasma ACE2 activity with a peak in concentrations between 1 and 2 weeks after disease onset, severity driven recruitment criteria (respiratory distress at emergency department) might have introduces a certain bias, as more severe cases have higher ACE2 at time of inclusion, with obviously having poorer outcomes. In other words, patients being included with a more severe disease manifestation ion day 0 (e.g. higher ACE2), may just be at another time-point in the course of COVID-19 disease, with an already pre-determined worse outcome. It would add a lot of valuer to the study, if the measurement time points would be related to the time of symptom onset, first positive test result or any other earlier time point that may would assure the direct comparability of disease severity groups.

Response: We agree with this comment. However, we are not able to get access to a longitudinal study of samples included at symptom onset or at the time of a positive test result. We have included the comment from the reviewer in the discussion because it is obviously a limitation to the study (page 22 line 16). We also mention that our study likely reflects clinical pratice were prognostic tests will not be carried our with no or mild symptoms. We would like to thank the Reviewer for bringing up this relevant point.

”Fourth, severity driven recruitment criteria (respiratory distress at emergency department) might have introduces a bias. Thus, patients with more severe COVID-19 at admission have both higher ACE2 at time of inclusion and a poorer prognosis. However, our study reflects clinical practice where laboratory tests are performed when patients are admitted to the hospital.”

The authors should further describe the control group of 78 Covid-negative patients described more clearly. Why did these patients at emergency units with respiratory distress?

Response: We agree with this comment. We included a description of the control group as below, while we are also referencing the original study where full patient characteristics are provided (page 7 line 5).

”COVID-19-negative subjects enrolled were older than COVID-19-positive patients, less Hispanic, and with greater baseline burden of chronic illnesses. Of the 78 COVID-19-negative subjects, 37 (47%) were diagnosed with non-COVID-19 pneumonia or acute lung injury (e.g., aspiration), 12 (15%) with congestive heart failure exacerbation, 6 (7.7%) with COPD exacerbation, 3 (3.8%) with acute pulmonary embolus, 11 (14%) with non-pulmonary sepsis or infection, and 8 (10%) with other illnesses. COVID-19-negative patients were significantly less inflamed than COVID-19-positive patients, median CRP 22 [IQR 9-67] versus 105 [IQR 48-161], p-value < 0.05, but illness acuity and outcomes were very similar between the two groups.”

Finally, the use of terminology within the whole paper is misleading, as the authors keep using the terms "ACE2 levels" and "ACE2 concentrations". Of note, the result of the proximal extension assay the authors used as a basis for their interpretation is given in the form of an artificial unit (NPX), that has been calculated by the manufacturer by arithmetically linking a series of Ct value based correction algorithms. Even the manufacturers point out on their webpage that the given readout cannot be compared to actual protein levels, which disables comparability to other studies. From a technical perspective, the big open question is why no calibration of the readout is performed to be able to provide actual protein levels. With a highly reproducible and standardized method as described by the manufacturers on their webpage, it should be easy to retrospectively include a valid calibration allowing for providing actual ACE2 concentrations instead of manufacturer invented units that prevent comparability with other studies. Moreover, a certain analytical validation vor ACE2 in the used PEA panel should be shown that compares the used "NPX" values with real concentration units or standard activity units in a defined set of clinical samples.

Response: We of course agree with these limitations. Unfortunately, many factors hinder us in generating actual protein concentrations. The manufacturer cannot generate protein levels from current data, we are not able to re-analyse the same samples in a different assay, and we are not in a position to collect samples from a validation cohort. We highlighted this in the limitations as described below (page 22 line 10). We have also reworded all mentions of ”levels” and ”concentrations” with ”plasma ACE2” or ”circulating ACE”.

“The plasma ACE2 levels were measured as relative protein concentrations using NPX (Normalized Protein eXpression) values. Therefore, it is not possible to determine a plasma ACE2 cut-off value to predict severe outcome or to compare findings in this study with results in studies measuring protein concentration or enzymatic activity.”

It should also be noted that reading the method section needs some revision as it currently reads like an advertisement for the used (commercially available) technology rather than an objective method description to be published in a research paper.

Response: We specifically referred to the website where the data is publicly available, although we agree that the methods should be tailored for scientists and we have therefore removed the unnecessary text about the technology. Thanks for this comment.

”Detailed description is available online (https://www.olink.com/mgh-covid-study/) (Fig 1) and has been published previously.[29] Briefly, the samples were analyzed by the Olink® Explore 1536 platform which includes measurement of the ACE2 protein. The Olink platform is based on Proximity Extension Assay (PEA) technology and has been validated previously.[31] Data generation consists of three main steps: normalization to known standard (extension control), log2-transformation, and level adjustment using the plate control. The generated data represent relative protein values, Normalized Protein eXpression (NPX), on a log2 scale where a larger number represents a higher protein level in the sample. For more information about Olink® Explore 1536, PEA and NPX, please visit www.olink.com.”

Reviewer #2: .This study investigated the association between plasma ACE2 levels and outcomes of COVID-19 patients, using clinical data and plasma samples from 306 COVID-19 positive and 76 COVID-19 negative patients. High baseline plasma ACE2 levels are reported to be associated with worse COVID-19 outcomes and patients with hypertension, pre-existing heart conditions or kidney disease had higher plasma ACE2 levels than those without.

Even as a marker I doubt very much that it will be useful for COVID-19 because the changes found in figure 2 are so small.

Response: We would like to thank the Reviewer for taking the time to help improving our manuscript. We really appreciate it. We agree with the comment that changes in ACE2 are small. Our study will require validation by others in assays designed to measure protein concentration or as mentioned further investigation with measuring ACE(1) as mentioned in the paper. We have reviewed our discussion and believe that the conclusions are balanced and in line with the reviewers comments, particularly as we have now included additional limitations of the study.

Main criticisms and suggestions for improvement:

- Overall, the paper is an excellent contribution but the authors should acknowledge the limitations of measurements of ACE2 in plasma where the levels are usually very low and even when mildly elevated in pathological conditions the significance remains uncertain. Specifically, it must be stated that ACE2 is a tissue enzyme and that the levels in the circulation are low in all species studied, including humans. Appropriate references should be given.

Response: Thank you for the comment. We completely agree that citing previous studies on measuring circulating ACE2 levels is a valuable adition. We have now added the following referneces and included a comment to highlight the limitation that ACE2 levels in circulations are low (page 4 line 22).

Epelman S, Shrestha K, Troughton RW, et al. Soluble angiotensin-converting enzyme 2 in human heart failure: relation with myocardial function and clinical outcomes. J Card Fail. 2009;15(7):565-571.

Tikellis C, Bialkowski K, Pete J, et al. ACE2 deficiency modifies renoprotection afforded by ACE inhibition in experimental diabetes. Diabetes. 2008;57(4):1018-1025.

Yamaleyeva LM, Gilliam-Davis S, Almeida I, Brosnihan KB, Lindsey SH, Chappell MC. Differential regulation of circulating and renal ACE2 and ACE in hypertensive mRen2.Lewis rats with early-onset diabetes. Am J Physiol Renal Physiol. 2012;302(11):F1374-1384.

”ACE2 is a tissue enzyme and circulation levels are low and the significance of measuring circulating ACE2 in pathologic conditions remains uncertain.”

Please acknowledge that the changes in COVID-19 are so small that they are not likely to be of any significance regarding the metabolism of the substrates of ACE2

Response: Thank you for bringing this to our attention. We did not consider this previously. It is an important perspective. We have mentioned this in the Discussion with the below wording (page 21 line 19):

”Importantly, changes in ACE2 levels in COVID-19 disease observed here are small and are likely not to be of any significance regarding the metabolism of the substrates of ACE2.”

- In the introduction the part on ACE2 receptor distribution must be modified to include the kidney as a main site of ACE2 . The authors do not seem aware of a critical important observation, namely that the expression of ACE2 in the lung is very low as shown by Serfozo et al using western blot and confirmed by others

See https://doi.org/10.1161/HYPERTENSIONAHA.119.14071 and https://doi.org/10.1038/s41467-020-19145-6

These references must be cited as well as others showing that the kidney is next to the intestine the organ that has the highest abundance of ACE2.

Response: We agree and have added the below sentence including the mentioned references (page 4 line 12).

- Perhaps the authors are not aware that ACE2 RNA levels do not necessarily imply protein levels. Actually, ACE2 can be post translationally regulated and be increased when the levels of mRNA are not. The authors may consult and cite a review on this by Lores et al. https://doi.org/10.1042/CS20200484

Response: We would like to thank the reviewer for underlining this, we have now included this important reference. However, the Olink assay measures protein, albeit with all the limitations already discussed above, therefore we did not measure RNA levels of ACE2.

- In the discussion it is stated that “ACE2 as a decoy receptor… is now being explored in COVID-19 disease.” Appropriate references should be given to be faithful to the literature. Perhaps they are not aware of these papers because they are recent but they should be cited. For instance, https://doi.org/10.1042/CS20200163 and https://doi.org/10.1681/ASN.2020101537

Response: Thank you very much for bringing these references to our attention. Indeed, we were not aware of this progress. We have included both citations.

- “A study performed on rats showed that the use of ACE inhibitors and/or use of AT1R-blocker led to an increased expression of ACE2”. It should be specified that in ref 34 ACE2 was only examined in cardiac tissue and not kidney or lung. Others found a decrease of kidney ACE2 after ACEi and ARBs and no effect on lungs. For instance, see https://doi.org/10.1681/ASN.2020050667

Response: Again we can only thank the reviewer for updating us and improving the manuscript. We have included this in the paper.

”Others found a decrease of kidney ACE2 expression and no effect on lung ACE2 expression with ACE-inhibitors and AT1R-blockers.” 

- “The population sampled on day 3 and day 7 therefore consists of patients with more severe disease compared with day 0.” Maybe ACE2 for day 3 and day 7 should be shown in a way that they are only compared to their matching samples of day 0?

Response: Thank you for this suggestion. We agree that this gives a nice representation of the data. We have included this in the Results section and shown data as Supplementary figure S1. The results are the same. We have therefore removed the above mentioned sentance in the Discussion.

6. PLOS authors have the option to publish the peer review history of their article (what does this mean?). If published, this will include your full peer review and any attached files.

Do you want your identity to be public for this peer review? For information about this choice, including consent withdrawal, please see our Privacy Policy.

Reviewer #1: No

Reviewer #2: No

---

## [Editor Report · Decision Letter 1]

24 May 2021

Plasma ACE2 predicts outcome of COVID-19 in hospitalized patients

PONE-D-21-06434R1

Dear Dr. Kragstrup,

We’re pleased to inform you that your manuscript has been judged scientifically suitable for publication and will be formally accepted for publication once it meets all outstanding technical requirements.

Kind regards,

Michael Bader

Academic Editor

PLOS ONE
---

## [Editor Report · Acceptance letter]

26 May 2021

PONE-D-21-06434R1 

Plasma ACE2 predicts outcome of COVID-19 in hospitalized patients 

Dear Dr. Kragstrup:

I'm pleased to inform you that your manuscript has been deemed suitable for publication in PLOS ONE. Congratulations! Your manuscript is now with our production department. 

Kind regards, 

on behalf of

Prof. Michael Bader 

Academic Editor

PLOS ONE